# Inhibitory Mechanism of Advanced Glycation End-Product Formation by Avenanthramides Derived from Oats through Scavenging the Intermediates

**DOI:** 10.3390/foods11121813

**Published:** 2022-06-20

**Authors:** Pei Zhu, Ying Zhang, Dianwei Zhang, Luxuan Han, Huilin Liu, Baoguo Sun

**Affiliations:** School of Food and Health, Beijing Technology and Business University, Beijing 100048, China; zp603541560@163.com (P.Z.); zhangying940907@163.com (Y.Z.); zhangdianwei@btbu.edu.cn (D.Z.); hlx19951112@163.com (L.H.); sunbg@btbu.edu.cn (B.S.)

**Keywords:** oat, avenanthramides, α-dicarbonyl compounds, inhibition, AGEs

## Abstract

As a special polyphenolic compound in oats, the physiological function of oat avenanthramides (AVAs) drives a variety of biological activities, and plays an important role in the prevention and treatment of common chronic diseases. In this study, the optimum extraction conditions and structural identification of AVAs from oats was studied. The inhibitory effect of AVAs from oats on advanced glycation end-products (AGEs) in a glucose–casein simulation system was evaluated, and this revealed dose-dependent inhibitory effects. The trapping capacity of AVAs to the α-dicarbonyl compounds of AGE intermediate products was determined by HPLC–MS/MS, and the results indicate that AVA 2c, AVA 2p, and AVA 2f exhibited the ability to capture *α*-dicarbonyl compounds. More importantly, AVA 2f was found to be more efficient than AVA 2p at inhibiting superoxide anion radical (O_2_^−^), hydroxyl radical (OH), and singlet oxygen (^1^O_2_) radical generation, which may be the main reason that AVA 2f was more efficient than AVA 2p in AGE inhibition. Thus, this research presents a promising application of AVAs from oats in inhibiting the food-borne AGEs formed in food processing.

## 1. Introduction

Advanced glycation end-products (AGEs) are produced by the nonenzymatic reaction of reducing sugars and protein amino groups [1]. It is generally accepted that the main compounds of AGEs are pathophysiologically derived from the ingestion of dietary N^ε^-carboxymethyllysine (CML), N^ε^-carboxyethyllysine (CEL), and pyrraline (PRL) [2]. Due to the potential health risks associated with AGEs, people are increasingly concerned about the level of food-borne AGEs [3,4,5]. Researchers from our laboratory summarized recent literature reports on CML contents in various foods, including milk, meat, fish, cereals, coffee, vegetables, and snacks [6]. One of the main sources of exogenous AGEs is food-borne AGEs, and approximately 10% of food-borne AGEs enter blood circulation. One-third of the food-borne AGEs are discharged through the kidney, but the other two-thirds accumulate in the body through the formation of covalent bonds between AGEs and tissues to induce a variety of diseases [7,8]. On the other hand, during food processing and oxidation of reducing sugars and proteins, many intermediate free radicals are produced [9]. Thus, inhibition of intermediates and free radicals is important in the formation of AGEs.

Oats are whole grains with high nutritional value that are rich in unsaturated fatty acids, *β*-glucan, and the anti-inflammatory polypeptide lunasin (approximately 0.197 mg/g), which can reduce the risk of chronic diseases such as cardiovascular disease, type II diabetes, and some types of cancers [10,11,12]. Avenanthramides (AVAs) are a special type of nitrophenol acid derivative in oats. They are a type of hydroxyl cinnamoyl aminobenzoic acid alkaloid [13], and various AVA structures can be derived from the *o*-aminobenzoic acid ring [14]. AVAs not only scavenge free radicals and have anti-wrinkling properties, but also have anti-inflammatory dermatitis properties. However, potential applications of AVAs in the inhibition or degradation of chronic diseases caused by AGEs have rarely been investigated [15].

In this study, AVAs extracted from oats were used as targets to inhibit food-borne AGEs associated with the Maillard reaction during food heating, processing, and storage (Figure 1). Three main AVAs from oats were identified by high-performance liquid chromatography (HPLC). The inhibitory effect of AVAs on AGEs formation was determined by a milk simulation system. The free radical and AGEs scavenging capacities of AVAs from oats for the α-dicarbonyl compounds glyoxal (GO), methylglyoxal (MGO), and 3-deoxyglucosone (3-DG) were studied by high-performance liquid chromatography–tandem mass spectrometry (HPLC–MS/MS) [16,17,18]. The inhibitory mechanism of AVAs on AGEs by the α-dicarbonyl compound pathway is discussed in detail. This study laid the theoretical foundation and technical guidance for improving the safety of food processing and promoting the healthy development of the food industry.

## 2. Materials and Methods

### 2.1. Chemicals and Materials

AVA 2c, AVA 2p, and AVA 2f (structure shown in Appendix A) were provided by Darwin Co., Ltd. (Beijing, China). In addition, *o*-phthalediamine (OPD), methanol (purity ≥99.5%), phosphate-buffered saline (PBS), aminoguanidine (AG), PRL, CML, CEL, 2,2-diphenyl-1-picryl hydrazyl (DPPH), and 2′-azino-bis(3-ethylbenzothiazoline-6-sulfonic acid) (ABTS·^+^) were also purchased from Darwin Co., Ltd. (Beijing, China). MGO, GO, and 3-DG were purchased from Pinxia Science and Technology Co., Ltd. (Beijing, China). Oats were provided by a local supermarket. The other reagents were purchased from J&K Scientific, Co., Ltd. (Beijing, China).

### 2.2. Optimization of the Extraction of AVAs from Oats

The concentration of methanol in the extraction solution (50%, 60%, 70%, 80%, 90%) and the ultrasonic time (10 min, 20 min, 30 min, 40 min, 50 min), number of extractions (2 times, 3 times, 4 times, 5 times, 6 times), and volume of extraction solution (5 mL, 10 mL, 15 mL, 20 mL, 25 mL) were optimized. Firstly, ammonium phosphate (0.023 g) was dissolved in methanol, and the pH was adjusted to 2.6 with phosphoric acid to serve as the extraction solution. Subsequently, 0.5 g of oat powder was taken as the sample and added to the extraction solution for ultrasonic treatment with different times. Then the treated samples were centrifuged at 10,000 rpm for 10 min in a high-speed refrigerated centrifuge (CR22N, Hitachi Machinery Co., Ltd., Tokyo, Japan). The supernatant was collected and the oat residues were extracted using the same method. Finally, all the supernatant was concentrated by rotary evaporator (R-210, Buchi Co., Ltd., Flawil, Switzerland) at 60 °C, redissolved with methanol, then detected and analyzed by HPLC (1260 infinity, Agilent Technologies Co., Ltd., Santa Clara, CA, USA).

### 2.3. HPLC Analysis of AVAs

The HPLC analysis referred to the method proposed by Skoglund, et al. [19], with certain modifications. The supernatant was filtered through an organic membrane filter (0.22 μm) before applying HPLC. The HPLC chromatographic conditions were set as follows: a reversed-phase C18 column (4.6 × 250 mm, 5 μm, Thermo Fisher Technologies Co., Ltd., Waltham, MA, USA) using a linear gradient. Two solvents were used for the mobile phase: (A) 0.1% formic acid, 5% acetonitrile, and 94.9% water; (B) 0.1% formic acid and 99.9% acetonitrile. Gradient elution was initiated at 95% in solvent A, decreasing linearly to 85% at 5 min, decreasing linearly to 70% at 35 min, remaining at 70% at 40 min, and returning to 95% at 50 min. The content of the AVAs was quantified at the UV wavelength of 340 nm. Injection volume was 10 μL, and the flow rate was 1.0 mL/min. The extracts of different samples and standard compounds were analyzed with the same conditions, and all of the above experiments were conducted in triplicate. Three main AVAs (AVA 2c, AVA 2p, and AVA 2f) were identified by comparing their retention times (RT) and UV spectra with those of standards. Next, quantitative data were calculated from their linear calibration curves. The results of the analyses for the three AVAs (AVA 2c, AVA 2p, and AVA 2f) are expressed in milligrams per kilogram DW (mg·kg^−1^ DW).

### 2.4. Antioxidant Activity Analysis of DPPH

The standard DPPH solution of 0.050 mg/mL concentration was obtained by accurately weighing 12.50 mg DPPH standard and dissolving it in anhydrous ethanol [20]. The solution was properly diluted before use to establish the absorbance of the DPPH solution as 0.70 ± 0.05. Then the DPPH solution (1.0 mL) and the different concentrations of extract solution (1 mL) were mixed for 30 min. Anhydrous ethanol was used as a blank control. The absorbance of each solution was determined at 517 nm by UV spectrophotometer (Cary 100 UV-Vis, Agilent Technologies Co., Ltd., Santa Clara, CA, USA). At the same time, 1 mL anhydrous ethanol was used to replace the sample for determination under the same conditions, the absorbance of which matched the initial absorbance. The experiment was measured three times in parallel, then the inhibition rate was calculated according to equation:(1)$$DPPH\;inhibition\;(\%)\;=\;[1\;-\;\frac{\left(A_{1}\;-\;A_{2}\right)}{A_{3}}]\;\times \;100$$
where *A*_1_ is the measured UV absorbance, *A*_2_ is the UV absorbance measured for anhydrous ethanol solution containing only DPPH, and *A*_3_ is the UV absorbance of the blank group.

### 2.5. Antioxidant Activity Analysis of ABTS·^+^

First, ABTS·^+^ stock solution (7.4 mmol/L, 0.2 mL) was mixed with K_2_S_2_O_8_ stock solution (2.6 mmol/L, 0.2 mL), and stood for 12 h under dark conditions at room temperature. After the reaction was completed, the mixed solution was diluted with PBS (pH = 7.4) until the absorbance value of the mixed solution was 0.7 ± 0.02 at the absorption wavelength of 734 nm, measured by a UV spectrophotometer (Cary 100 UV-Vis, Agilent Technologies Co., Ltd., Santa Clara, CA, USA). The final mixed solution was then obtained, and we regarded the ABTS·^+^ as the working solution. [21]. Then, the extract solution (0.2 mL) was added to the ABTS·^+^ working solution (0.8 mL), and the solution was vibrated for 10 s by vortex oscillator and allowed to stand for 6 min. The absorbance value was measured at 734 nm. The control group of 95% anhydrous ethanol (0.2 mL) was added to the ABTS·^+^ working solution (0.8 mL). The formula for the determination of antioxidant capacity by the ABTS·^+^ method was as follows:(2)$${ABTS\cdot }^{+}\;inhibition\;(\%)\;=\;\frac{\left(A_{0}\;-\;A\right)}{A_{0}}\;\times \;100$$
where *A* is the UV absorbance at 734 nm, *A*_0_ is the UV absorbance of 95% anhydrous ethanol measured at 734 nm.

### 2.6. Construction of the Glucose–Casein Simulation System

The glucose–casein simulation system was constructed according to Nguyen et al., with some modifications [22]. Firstly, 30 g of casein and 27 g of D-glucose were dissolved in 1 L PBS. The solution was stirred at room temperature for 8 h until the casein was completely dissolved. Following this, 6 mL of the prepared solution was placed into the reaction flask and reacted at 60 °C or 65 °C in a water bath for 30 min. Then, 1.5 mL of AVA extract was added into the 1.5 mL glucose–casein simulation system and heated at 60 °C or 65 °C for 30 min. This process was repeated for the blank group and the positive control group, with 1.5 mL distilled water and 1.5 mL AG replacing the AVA extract, respectively.

### 2.7. Determination of the Contents of Main AGEs

The sample pretreatment referred to the method of Assar et al., with certain modifications [23]. AVA extract equivalent to 2 mg of protein was added to sodium borate buffer solution (0.5 mol/L, pH = 9.2) to a final concentration of 0.2 mol/L. Then, sodium borohydride (2 mol/L, prepared with 0.1 mol/L NaOH) was added, with a final concentration of 0.1 mol/L, to reduce the extract at 4 °C for 10 h. The obtained solution was supplemented with 60% trichloroacetic acid, so that the final concentration of trichloroacetic acid was 20%. The protein was precipitated by centrifugation in a high-speed centrifuge at 10,000 rpm for 10 min. The obtained protein was washed twice with 100% acetone, and 1 mL of 6 mol/L HCl was added, followed by hydrolysis at 110 °C for 24 h. The hydrolyzate was then blow-dried with a nitrogen blower (Ugc-24m, Beijing Yousheng United Technology Co., Ltd., Beijing, China) and redissolved in 2 mL of ultrapure water. The redissolved hydrolyzates were removed onto a solid phase extraction column; the PCX SPE columns were activated with 3 mL of methanol and 3 mL of water successively, then, 2 mL of hydrolyzate was passed through the column, which was subsequently washed with 3 mL of water and 3 mL of methanol, successively. Finally, the SPE columns were eluted with 5 mL of methanol solution containing 5% ammonia. After collecting the eluents, they were blow-dried with nitrogen, redissolved in 2 mL of ultrapure water, and stored at 4 °C. Then we used high-performance liquid chromatography–mass spectrometry (HPLC–MS/MS, 1290 Infinity, Agilent Co., Ltd., Santa Clara, CA, USA) to determine the AGE content of each group, thus conducting three parallel experiments.

#### 2.7.1. The HPLC–MS/MS Conditions for CML and CEL

The HPLC–MS/MS analysis for CML and CEL was performed according to the method reported by Assar et al. [23]. Using a chromatographic column, Inertsil ODS-C18 (150 × 4.6 mm, 4.6 μm), the following parameters were applied: flow rate, 0.2 mL/min; sample volume, 10 μL; mobile phase A, 0.1% TFA (trifluoroacetic acid) aqueous solution; mobile phase B, acetonitrile; analysis time, 25 min; gradient elution procedure, mobile phase A was 40–10% and 0.5–4 min, mobile phase B was 10–40% and 0–0.5 min. The MS/MS conditions were as follows: monitoring mode, MRM; mass spectrometry module, triple quadrupole tandem mass spectrometer; ion mode, ESI (+); ion source temperature, 300 °C; taper hole voltage, 15 psi; capillary voltage, 4 kV; and setting of the MRM mode, *m/z* 205.0 to 84.2 or 130.0, and *m/z* 219.0 to 84.2 or 130.0.

#### 2.7.2. The HPLC Conditions for the Determination of the PRL

Inertsil ODS-SP column (4.6 mm × 250 mm, 5 μm); mobile phase A, water (0.1% TFA); mobile phase B, acetonitrile-water (1:1, *v/v*); gradient elution procedure, where the content of mobile phase B was 0%, 15%, 20%, 100%, and 0% at 10, 30, 35, 40, and 45 min, respectively; analysis time, 45 min; flow rate, 0.4, 0.6, 0.8, 1.0, and 1.2 mL/min; sample volume, 5 μL; column temperature, room temperature; and UV-visible light detection wavelength, 298 nm.

### 2.8. Determination of AGEs Intermediates Captured by AVAs

Firstly, 1 mL of 1 mM AVA 2p, AVA 2c, and AVA 2f and 1 mL of 1 mM MGO, GO, or 3-DG were reacted at 190 °C in an oil bath for 10 min, respectively. Then they were removed and cooled immediately, and 2 mL of 20 mM OPD was added for 30 min, to be subsequently tested. Before detection by HPLC, we filtered the prepared solution with a 0.22 μm organic filter membrane. The HPLC conditions were as follows: ODS C18 (150 mm × 4.6 mm, 5 μm); mobile phase A:B = MeOH:H_2_O = 50:50; flow rate, 0.6 mL/min; sample volume, 5 μL; and detection wavelength, 313 nm. The MS/MS conditions were as follows: full sweep mode; scanning range, *m/z* = 100−1000; ion source temperature, 300 °C. Then, the clearance rate of AGEs intermediates was calculated as follows:
(3)$$Clearance\;(\%)\;=\;\frac{\left(A_{0}\;-\;A\right)}{A_{0}}\;\times \;100$$
where *A*_0_ is the concentration of quinoxaline without AVAs, and *A* is the concentration of quinoxaline after adding AVAs.

### 2.9. Experiments on Scavenging Free Radicals by AVAs

For hydroxyl (OH) radical scavenging activity, 200 μL of FeSO_4_ solution (5 mg/mL), 20 μL of DMPO, 160 μL of deionized water, and 20 μL of H_2_O_2_ (30%) were mixed and further reacted for 5 min. Then, samples were taken for testing, and used as controls. For the other samples, 160 μL of deionized water was replaced by 160 μL of sample solution. For singlet oxygen (^1^O_2_) radical scavenging activity, firstly, PBS buffer was used as the configuration solvent. Then 100 μL of xanthine solution (10 mM), 100 μL of (1 U/mL) xanthine oxidase, 20 μL of DMPO, and 180 μL of PBS buffer were mixed and further reacted for 10 min. The mixture was taken for testing and used as a control. For the other samples, 180 μL of deionized water was replaced by 180 μL of sample solution. For superoxide anion radical (O_2_^–^) scavenging activity, PBS buffer was used as the configuration solvent to configure 1 mg/mL of TiO_2_ aqueous solution. Then, 200 μL of (1 mg/mL) TiO_2_ aqueous solution, 200 μL of TEMP (50 mM), and 100 μL of deionized water were mixed, and samples were loaded and tested by capillary electrophoresis. Once illuminated by a 300 W xenon lamp for 10 min, data were collected to represent the control group. For the other samples, 100 μL of deionized water was replaced by 100 μL of sample solution.

### 2.10. Statistical Analysis

All experiments were conducted in triplicate unless indicated otherwise, and the results are expressed as the mean ± standard deviation (SD). The statistical differences were analyzed by one-way ANOVA using SPSS software version 17.0. Values of *p* < 0.05 were considered statistically significant. The figures were drawn using Origin 2021 software.

## 3. Results and Discussion

### 3.1. Optimization Results of AVA Extraction from Oats

The conditions of AVA extraction were first optimized by exploring the effect of the ratio of methanol in the extraction solution, the number of extractions, ultrasonic time, and the volume of extraction solution on the content of extracted AVAs from oats, as shown in Figure 1. When the extraction solution was 70% methanol (100 mL), the average extracted AVA content reached a maximum of 193 μg/g; when the extraction solution was 50% methanol, 60% methanol, 80% methanol, and 90% methanol, the average extracted AVA contents were 154, 160, 188, and 179 μg/g, respectively (Figure 1A). Therefore, with the increase in methanol percentage in the extraction solution, the extracted AVA content increased initially, and then decreased; this might be caused by the excess methanol, as it can also extract other impurities, resulting in the extracted AVA content decreasing. Thus, 70% methanol was selected as the optimal extraction solution. As shown in Figure 1B, the AVA content was 201 μg/g when ultrasonically extracted for 30 min, which was higher than when extracted for 10, 20, 40, and 50 min, with contents of 154, 168, 179, and 173 μg/g, respectively. This might be attributed to the degradation of AVAs with the increased temperature caused by ultrasonication. Therefore, 30 min was selected as the optimal ultrasonication time. As for the number of extractions, the AVA content reached a maximum of 260 μg/g when extracted three times; beyond this, with the number of extraction times increasing, AVAs extracted from oats decreased, probably because excess extraction times can cause the level of residual supernatant fluid, as well as the spin steamed area, to increase, resulting in partial loss of AVAs. Therefore, three extractions were chosen as the optimal number of extractions (Figure 1C). When the solid–liquid ratio was 1:10, the content of AVAs in oats reached a maximum of 267 μg/g. With an increase in the volume of extraction liquid, the extraction amount of AVAs in oats decreased, but was relatively smooth. This might be because of the fact that an increase in the volume of extraction liquid can extract other substances at the same time. Hence, a solid–liquid ratio of 1:10 was selected as the optimal extraction solid–liquid ratio (Figure 1D). The optimized conditions could effectively extract AVAs from oats were, therefore: 70% methanol, 10 mL of the extraction volume, 30 min of extraction time, and three rounds of extraction.

### 3.2. Identification of the AVAs from Oat Extracts

In this study, the AVAs of the oat extracts were further identified by HPLC, and the results are shown in Appendix A. Three peaks were observed and identified as AVA 2c, AVA 2p, and AVA 2f, with RTs of 2.087 min (*m/z* = 315.9), 3.195 min (*m/z* = 299.9), and 3.421 min (*m/z* = 329.9), respectively. This result indicates that the AVAs are the main components of oat extracts. In addition, the contents of AVA 2c, AVA 2p, and AVA 2f were determined to be 45 ± 1.12, 103 ± 1.62, and 196 ± 2.30 μg/g, respectively; thus, the content of AVA 2f in oat extracts was the highest. These results demonstrate that these three types of AVAs are the main components of oat extracts, and AVA 2f had the highest content.

### 3.3. Evaluation of the Antioxidant Activity of Oat Extracts

The antioxidant activity of oat extracts was investigated by assessing their DPPH and ABTS·^+^ scavenging activities. The scavenging activity of DPPH free radicals was 47.6%, and the scavenging activity of ABTS·^+^ free radicals reached 53.2% (Figure 2A). These results show that the oat extracts had good scavenging antioxidant activity, which may be attributed to the alkaloids in oat extract.

### 3.4. Inhibition of AVAs Extracted from Oats on AGEs in the Simulation System

The standard curves of CML, CEL, and PRL are shown in Appendix A. The results show that, under the optimal extraction conditions, the inhibitory rate of AVAs against CML was as high as 51.2%, while the inhibitory rate of AG (aminoguanidine, positive control, effective inhibitor of CML) against CML was as high as 53.6% (Figure 2B). It could be concluded that the AVAs extracted from oats had a good inhibitory effect on AGEs. Similarly, the inhibitory rate of AVAs against CEL was as high as 40.8%, while the inhibitory rate of AG against CEL was as high as 51.6% (Figure 2C). The inhibition rate of the AVAs from oat extract against PRL was 39.5% (Figure 2D), and that of AG against PRL was 52.9%. Therefore, AVAs extracted from oats had the highest inhibitory rate against CML. Studies on the inhibition of AGE formation in recent years are summarized in Table 1. As can be seen from Table 1, the components of inhibitors, with diverse sources, were mainly phenolic compounds, and the inhibition capacity of AVAs from oats is comparable with, or even better than, that reported in an earlier study, which demonstrated the possibility of AVAs from oats as AGE inhibitors.

**Table 1 foods-11-01813-t001:** Summary of studies on AGE inhibition by different inhibitors.

Target	Inhibitors		Inhibition (%)	References
	Source	Main Components		
PRL	Highland barley whole grain	Phenolic compounds	52.03	[17]
CML	Standard substance	Phenolic compounds	31.77	[24]
PRL	Highland barley vinasse	Phenolic compounds	49.22	[17]
CML	Lotus seed waste	B-type phenolic acids	29.40	[25]
CML	Highland barley bran	Phenolic acids	45.58	[26]
AGEs	Millet	Phenolics	68.3	[3]
AGEs	Olive mill wastewater	Polyphenols	43.0	[27]
CML	Green coffee	Chlorogenic acids	64.5	[28]
AGEs	Standard substance	Capsaicin	60.0	[29]
CML	Standard substance	Catechins	-	[30]
CEL, CEL, PRL	Oats	AVAs	39.5–51.2	This Work

### 3.5. Experimental Results of Capturing α-Dicarbonyl Compounds by AVAs

As previously stated, α-dicarbonyl compounds are considered an important intermediate product used to form AGEs [31]; therefore, the abilities of three types of AVAs to capture α-dicarbonyl compounds (GO, MGO, and 3-DG) were investigated, and the structures of formed adduct products were analyzed by HPLC–MS/MS. AVAs have been indicated to be a group of phenolic alkaloids, and can effectively capture α-dicarbonyl compounds through an addition reaction, which is the main mechanism of inhibiting AGE formation [32]. Lo et al. [33] showed that the C_8_ site of EGCG is susceptible to nucleophilic attack by MGO. A similar study performed by Lv et al. [34] investigated the ability of the active component tetrahydroxy tilbene glycoside (THSG) to capture MGO, and their results show that the C_4_ and C_6_ sites of the THSG phenolic ring are easily replaced by MGO to generate single MGO adducts and double MGO adducts. High-resolution mass spectra in Figure 3a–e and Figure 4A indicated the successful adduction of AVA 2p and MGO, as well as 2MGO, GO, 2GO, and 3-DG, resulting in the formation of corresponding mono- and diadducts through the replacement of hydroxyl groups and/or the carboxyl group in the benzene ring of AVA 2p by *α*-dicarbonyl compounds. It should be noted that all three *α*-dicarbonyl compounds (MGO, GO, and 3-DG) can react with AVA 2p to form monoadducts, while only MGO and GO can form the diadducts with AVA 2p. We tried to explore the reaction between AVAs 2p and 3MGO, 3GO, 3-DG, but none of the three triadducts were detected. Similar results were observed for AVA 2c and AVA 2f, as presented in Figure 3f–o. Specific synthetic routes are shown in Figure 4B,C, respectively. These synthetic addition products proved that AVAs captured the intermediate products in the formation of AGEs (*α*-dicarbonyl compounds), and the addition of AVAs and *α*-dicarbonyl compounds blocked the process of generating AGEs, thus inhibiting their formation.

### 3.6. Clearance Rate Results

In order to further determine the inhibition mechanism, the clearance rates of α-dicarbonyl compounds by three types of AVAs were evaluated by HPLC–MS/MS by transforming α-dicarbonyl compounds into quinoxaline with UV absorption. MGO, GO, and 3-DG are the key intermediate products in the formation of CEL, CML, and PRL, respectively. The higher the clearance of the intermediate product, the better the effect of inhibition by AVAs [16]. As shown in Appendix A, the scavenging rates of three main alkaloids, AVA 2c, AVA 2p, and AVA 2f, on MGO were 65.5%, 44.0%, and 71.2%, respectively, which demonstrates that the inhibition of CEL by AVAs was mainly attributed to the capturing of intermediate product MGO. The scavenging efficiency of AVA 2p, AVA 2c, and AVA 2f for GO was calculated by the amount of remaining GO. The data presented in Appendix A show that most GO was cleared by AVA 2p, AVA 2c, and AVA 2f, with clearance rates of 62.8%, 72.7%, and 79.2%, respectively; therefore, although there were some other pathways of CML inhibition, including active carbonyl group capture, antioxidant activity, glucose oxidation inhibition, and amino acid binding inhibition/competition, the reaction of GO with AVAs was still the main mechanism that inhibited CML formation. The scavenging rates of the three main alkaloids on 3-DG were 45.7%, 50.9%, and 80.4%, respectively. It should be noted that 3-DG can only form monoadducts with AVAs, compared with MGO and GO, indicating that the inhibition of PRL might not only occur by scavenging 3-DG, but also by some other pathways. All these observations confirm that the inhibition mechanism of AVAs on the formation of AGEs was mainly assigned to the trapping of α-dicarbonyl compounds by adduction reaction with AVAs.

### 3.7. Scavenging of Radicals in the Glucose–Casein Simulation System

As an intermediate of the oxidation reaction, free radicals in food could be used to judge the degree of food oxidation and predict the quality of food by ESR spectroscopy analysis. The oxidation processes of CML, CEL, and PRL as reducing sugars and proteins also gained plenty of intermediate free radicals in the formation process of compounds. Free radicals can mediate the transformation of Amadori products into AGEs. Therefore, the formation of AGEs, and the process of glycosylation, can be analyzed by the ESR technique. The ability of AVA 2f, AVA 2p, and AVA 2c to inhibit the formation of radicals was performed in simulation systems of lysine and glucose. AVA 2p, AVA 2f, and AVA 2c exhibited excellent scavenging activity for free radicals (Figure 5A–C). AVAs were found to be efficient inhibitors of ·O_2_^–^, ^1^O_2_, and ·OH radicals, which might be the main reason that AVAs were excellent for AGE inhibition. AVA 2p, AVA 2c, and AVA 2f scavenge free radicals mainly through their B rings. AVAs have hydrogen atoms on the B ring phenolic hydroxyl group, which act as free radical receptors, quenching free radicals and blocking chain oxidation reactions.

## 4. Conclusions

In this study, we utilized in vitro Maillard reaction systems to explore the inhibitory effect of AVA extracts on AGEs in a food model system of glucose–casein. Through the detection of CML, CEL, and PRL, the inhibition effect of AVAs on food-borne AGEs was 51.2%, 40.8%, and 39.5%, respectively. The inhibitory mechanisms were confirmed by the effective inhibition by the AVAs of the intermediates MGO, GO, and 3-DG. This was due to their ability to scavenge active carbonyls by forming adducts that block the carbonyl compound from reacting further to make AGEs. The AVA–MGO, AVA–GO and AVA–3DG adducts were determined by HPLC–MS/MS. The reaction mechanism of AVAs for removing intermediates was confirmed. Meanwhile, the free radicals in the food model system were detected by ESR, and it was found that the free radical content was significantly reduced after the addition of AVAs, which also explained why AVAs had a strong scavenging effect on the free radicals in the system. The antioxidant capacity of AVAs was also efficient, at 46.6% and 53.2% with the DPPH and ABTS·^+^ methods, respectively. During simulated food processing, AVA extracts, a special class of phenolic acid derivatives containing nitrogen in oats, might inhibit the formation of food-borne AGEs by inhibiting the adduction of protein and α-dicarbonyl compounds, and inhibiting oxidative stress. This study presents a promising strategy for inhibiting the food-borne AGEs formed in food processing.

## Data Availability

Data is contained within the article or Appendix A.

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
