# Peer review of "Inhibitory Mechanism of Advanced Glycation End-Product Formation by Avenanthramides Derived from Oats through Scavenging the Intermediates"

_foods, 2022, doi:10.3390/foods11121813_

Round 1
Reviewer 1 Report
The manuscript entitled "Inhibitory mechanism of advanced glycation end-product for-2 mation by avenanthramides derived from oats through scavenging the intermediates" investigates an intriguing subject with relevant impact on food composition and health, regarding the formation of advenced glycation end products (AGEs) and their potential inhibition. Although the generation of AGEs in vivo represents a larger issue for human health, the ability to reduce AGEs formation in the food matrices is still an interesting aspect to consider. Tha manuscript is well written.
Minor:
Line 248: "As shown in Figure S2, Theses pictures..." explain or rephrase
Reviewer 2 Report
Very interesting study about oats' bioactive components- avenanthramides and their relation to AGEs inhibitory activity. However, the investigations seem interesting in the beginning, but the rest of the manuscript doesn't prove the high quality of this paper. Please find some comments that may be to improve the text.
I would rather avoid in the abstract usage of 2c, 2p etc.
"·O2 – , ·OH and 1O2" should be edited.
"in vivo" should be written in italics. Line 33 how in vivo AGEs can be food-borne? there are endogenous AGEs which are glycated proteins in human tissues, in vivo; and exogenous AGES which are also named dietary AGEs that come from food, and diet. Food-borne AGEs are that formed in the food via different processing e.g. Maillard reaction.
Abbreviations (general comment): if the name is used once there is no need to write an abbreviation e.g. CVD.
Line 43 "some types of cancers".
Line 53: what simulated food system was used and why?
Line 59: "national nutrition" please add any data about the consumption of oats in different countries and the processing of oat. According to the methodology and description, the aim of the research is not reached.
Methodology: what means: 2c, 2f, and 2p.
"the o-ami- 45 nobenzoic acid ring" please write "o-" using italics.
Lie 46: "combined" is not necessary here.
Line 72: "times" and "material ratio" is not well-defined names, Authors should find other expressions.
2.2 could be presented on the flowchart. One's Authors wrote Japan, the other time JPN. Moreover, the information about spectrophotometer equipment and producer has not been mentioned in DPPH and ABTS methods.
Line 111: what was the absorbance value?
Line 151: any references?
Figure 1: twice the same 'time'.
why Table 1 is presented in the results and discussion part?
Reviewer 3 Report
The study is well organized, however language is compromised at some parts of the manuscript.
Provide a reference for "Construction of the glucose-casein simulation system"
Was there any correlation between antioxidant activity and Inhibition of AVAs by the extract??
Discussion section should be improved and more reasons should be provided to support the AVAs inhibition ability and possible mechanism should be discussed briefly.
Figure quality and presentation should be improved.
